# Simultaneous Administration of Recombinant Measles Viruses Expressing Respiratory Syncytial Virus Fusion (F) and Nucleo (N) Proteins Induced Humoral and Cellular Immune Responses in Cotton Rats

**DOI:** 10.3390/vaccines7010027

**Published:** 2019-03-04

**Authors:** Yoshiaki Yamaji, Akihito Sawada, Yosuke Yasui, Takashi Ito, Tetsuo Nakayama

**Affiliations:** 1Laboratory of Viral Infection II, Kitasato Institute for Life Sciences, Kitasato University, Tokyo 108-8641, Japan; di12004@st.kitasato-u.ac.jp (Y.Y.); akihito@lisci.kitasato-u.ac.jp (A.S.); itot@lisci.kitasato-u.ac.jp (T.I.); 2Health Center, Keio University, Kanagawa 223-8521, Japan; yyasui@z5.keio.jp

**Keywords:** RSV, F, NP, cotton rat, recombinant virus

## Abstract

We previously reported that recombinant measles virus expressing the respiratory syncytial virus (RSV) fusion protein (F), MVAIK/RSV/F, induced neutralizing antibodies against RSV, and those expressing RSV-NP (MVAIK/RSV/NP) and M2-1 (MVAIK/RSV/M2-1) induced RSV-specific CD8^+^/IFN-γ^+^ cells, but not neutralizing antibodies. In the present study, MVAIK/RSV/F and MVAIK/RSV/NP were simultaneously administered to cotton rats and immune responses and protective effects were compared with MVAIK/RSV/F alone. Sufficient neutralizing antibodies against RSV and RSV-specific CD8^+^/IFN-γ^+^ cells were observed after re-immunization with simultaneous administration. After the RSV challenge, CD8^+^/IFN-γ^+^ increased in spleen cells obtained from the simultaneous immunization group in response to F and NP peptides. Higher numbers of CD8^+^/IFN-γ^+^ and CD4^+^/IFN-γ^+^ cells were detected in lung tissues from the simultaneous immunization group after the RSV challenge. No detectable RSV was recovered from lung homogenates in the immunized groups. Mild inflammatory reactions with the thickening of broncho-epithelial cells and the infiltration of inflammatory cells were observed in lung tissues obtained from cotton rats immunized with MVAIK/RSV/F alone after the RSV challenge. No inflammatory responses were observed after the RSV challenge in the simultaneous immunization groups. The present results indicate that combined administration with MVAIK/RSV/F and MVAIK/RSV/NP induces humoral and cellular immune responses and shows effective protection against RSV, suggesting the importance of cellular immunity.

## 1. Introduction

Respiratory syncytial virus (RSV) is a common cause of viral lower respiratory tract infections and causes serious diseases in infants, the elderly, and immunocompromised patients [1]. More than 70% of children are infected with RSV in the first year of their lives and nearly 100% by two years of age. RSV causes hospital infections and serious clinical conditions in infants, even though they have high titers of maternal antibodies against RSV [2]. Infants acquire RSV-specific IgG and IgA antibodies and cytotoxic lymphocytes (CTL) after repetitive natural RSV infections [3,4]. The presence of RSV-specific antibodies has been correlated with protection against RSV infection in mice [5].

The formalin-inactivated RSV (FI-RSV) vaccine was developed in the mid-1960s. RSV-naïve infants who received the FI-RSV vaccine developed moderate or severe lower respiratory tract diseases when they were naturally exposed to RSV, whereas those who received an inactivated parainfluenza type 3 virus vaccine as a control did not [6,7]. This phenomenon may be explained in part by formalin disrupting epitopes on RSV-fusion protein (F) and the FI-RSV vaccine did not induce effective neutralizing antibodies (NA) after the vaccination [6,7,8,9,10,11]. The underlying immune mechanisms have been proposed. Th2-skewed immune responses were shown to be induced by the FI-RSV vaccine, and eosinophilia occurred after natural RSV infection and induced poor CTL responses [11]. Previous studies reported that RSV vaccine-enhanced Th2 responses were regulated by the induction of Th1-type immune responses [12,13,14]. Marked inflammation was inhibited by cytokines, including IL-10, 12, 17, and 27 and IFN-γ [15,16,17]. These data were based on experiments on mice and cotton rats. A critical point in RSV vaccine development is the induction of balanced humoral and CTL responses that protect the host from severe RSV infections.

RSV-F and G proteins have been shown to induce both humoral and cellular immunity in a mouse model, while the M2-1 protein induces cellular immunity. The CD8+ T cells responding to F85–93 and F352–360 peptides were infiltrated in the lungs in mouse immunized with recombinant vaccinia virus expressing RSV F protein after RSV infection [18]. RSV F85–93 peptide region induced CD8 T cell response in mice immunized with influenza virus-vectored virus expressing RSV F85–93 [19]. The Th1-type responses induced by RSV-F were insufficient to regulate inflammatory responses after RSV infection. Therefore, nucleoprotein (NP) or M2-1 is considered one of the effective proteins inducing CTL activities in mice [20,21].

A live-attenuated virus vaccine can induce both humoral and cellular immune responses but is not developed for RSV. Several approaches have been proposed, such as replication-defective vectors or chimeric live vectored-vaccine candidates, using parainfluenza virus, Sendai virus, small pox vaccine, and BCG [22]. Most vector viruses, except small pox, had no experience of clinical use in human. The AIK-C measles vaccine strain has been used as National Immunization Program in Japan since 1978 and the safety and immunogenicity are established. Infectious cDNA clone was constructed from the AIK-C seed virus by reverse genetics and the development of recombinant vectored-vaccines derived from licensed vaccine strain is considered theoretically safe and effective [23]. A live measles vaccine-based vector system has been developed and we also produced the recombinant measles viruses MVAIK/RSV/F, MVAIK/RSV/M2-1, and MVAIK/RSV/NP expressing RSV-F, M2-1, and NP [23,24,25]. MVAIK/RSV/F induced effective NA in cotton rats and significantly reduced the recovery of infectious RSV in the lungs after the challenge. However, the infiltration of inflammatory cells was observed in the peri-bronchial space. MVAIK/RSV/M2-1 and MVAIK/RSV/NP induced RSV-specific CTL activities in cotton rats and reduced inflammatory responses after the RSV challenge. We hypothesize that balanced immune responses may be induced using these two proteins and may control inflammation. In the present study, we demonstrated sufficient protective immunity using simultaneous administration with MVAIK/RSV/F and MVAIK/RSV/NP in the cotton rat model. 

## 2. Materials and Methods

### 2.1. Viruses and Cells

Full length infectious cDNA was constructed from the AIK-C live attenuated measles vaccine strain, and infectious virus (MVAIK) was recovered. The *Asc* I restriction site was artificially introduced at the P/M junction and heterologous genes were inserted using the *Asc* I restriction site [23]. The full-length F gene of wild-type RSV was cloned at the P/M junction and infectious recombinant virus (MVAIK/RSV/F) was recovered [24]. The RSV-M2-1 and NP genomes were cloned from RSV subgroup A wild type and inserted at the P/M junction of the measles virus vector. Recombinant measles viruses (MVAIK/RSV/M2-1 and MVAIK/RSV/NP) were recovered from B95a cells. Virus growth of the recombinant viruses was examined. Culture fluids were obtained on Days 1, 3, 5, and 7 of the culture and infectivity was shown as 10^n^ TCID_50_/mL in Vero cells. RSV (Long strain) was propagated in HEp-2 cells for the challenge virus. The expression of the F protein was mentioned in the previous report [24] and that of the NP and M2-1 was also reported [25]. 

B95a cells were maintained in RPMI-1640 medium (Sigma-Aldrich, Dorset, UK) supplemented with 10% FCS. HEp-2 cells were maintained in MEM (Sigma-Aldrich, Dorset, UK) supplemented with 10% FCS.

### 2.2. Animals and Immunization

The experimental scheme is shown in Figure 1. Seven- to eight-week-old female inbred cotton rats (three or four animals per group) were immunized by an intramuscular injection with 10^6^ or 5 × 10^5^ TCID_50_ of MVAIK/RSV/F (showing F 10^^6^ and F 10^^5^ groups). Cotton rats were immunized with a total of 5 × 10^5^ TCID_50_ of MVAIK/RSV/F and 5 × 10^5^ TCID_50_ of MVAIK/RSV/NP in the group F+NP, and those immunized with 5 × 10^5^ TCID_50_ of MVAIK/RSV/F and MVAIK/RSV/M2-1 in the F+M2-1 group. In the F 10^^6^, F+NP, and F+M2-1 groups, vaccination dosages were adjusted to the same infectivity of measles virus. Cotton rats were re-immunized on Day 56. Sera were obtained for the detection of serological responses immediately before and on Days 21, 35, 56, and 84. Cotton rats were anesthetized and infected with 1 × 10^6^ PFU of RSV subgroup A (Long strain) in a volume of 0.5 mL through an intranasal route and were sacrificed on Day 88. The spleen, bronchoalveolar lavage (BAL), and lung tissues were collected for the detection of cellular immune responses. Rats non-immunized (naïve) and immunized with MVAIK (empty vector of MVAIK: AIK-C group) were used as control.

Animal experiments and study protocols were approved by the Animal Ethics Committee of Kitasato University, Kitasato Institute for Life Sciences (Protocols No. 17-027).

### 2.3. Intracellular Cytokine Staining (ICS)

ICS was performed using the Cytofix/cytoperm kit (BD Pharmingen, San Diego, CA, USA) according to the manufacturer’s instructions. A total of 1 × 10^6^ freshly isolated splenocytes were stimulated with 1 μM of an individual peptide or inactivated virus in the presence of 1 μg/mL brefeldin A (Sigma-Aldrich, Dorset, UK) in 1000 μL of culture medium at 37 °C for five hours [12]. Cultures were stimulated with UV-inactivated RSV Long strain, F, NP, and M2-1 peptides as shown in Table 1. F peptides are recognition sites of palivizumab (synagis®). NP_306–314_ is based on the human experiments and the others are on mouse. After the stimulation, splenocytes were washed and incubated with an anti-CD8 antibody (R&D Systems, Minneapolis, MN, USA) at 4 °C for 30 min. Splenocytes were fixed with fixation buffer in the Cytofix/cytoperm kit, and intracellular cytokines were stained with a goat IgG antibody against IFN-γ (R&D Systems, USA) and anti-goat IgG PE-Cy7 (Santa Cruz Biotechnology, Inc., Dallas, TX, USA) at 4 °C for 60 min. Cellular populations were analyzed using flow cytometry with *Cytomics* FC 500 (Beckman Coulter, Inc., Indianapolis, IN, USA) and counted until 100,000 cells. 

### 2.4. Serology

The neutralization test against RSV was performed with the 50% plaque reduction assay using the Long strain [24,25]. Briefly, serum samples were serially diluted by four-fold, starting from a 1:10 dilution, and mixed with an equal volume of RSV (100 PFU) in MEM at room temperature for 1 h. These mixtures were used to inoculate a monolayer of HEp-2 cells in a 24-well plate. Plates were incubated at 37 °C for 1 h in 5% CO_2_ and then overlaid with MEM supplemented with antibiotics, 5% fetal bovine serum (FBS), and 0.5% agar. After being incubated at 37 °C for six days in 5% CO_2_, cells were fixed with 1% formaldehyde. Agar was removed, and cells were stained with neutral red. Plaque numbers were counted and NA titers were calculated as the reciprocal of serum dilutions showing a 50% reduction in the plaque number. NA titers are expressed as 2^n^.

Antibodies against the measles virus were measured using a particle agglutination kit (Serodia®-Measles, Fuji Rebio, Tokyo, Japan) in order to detect the agglutination of gelatin particles. Gelatin particles are coated with purified measles virus particles. Sera were serially diluted by two-fold, starting from a 1:10 dilution, and each serum dilution was mixed with an equal volume of gelatin particles, according to the recommendation of the manufacturer. PA antibody titers were expressed as the reciprocal of the serum dilutions that induced particle agglutination.

RSV antigen was prepared from the culture fluid of Vero cells infected with the Long strain and were roughly purified through centrifugation by 6000 rpm to remove cellular debris, containing 10^7^ PFU/mL. They were coated in 96-well ELISA plates at 4 °C overnight. A 200-fold serum dilution was added to the wells, which were then incubated for 1 h. After the incubation, plates were washed three times with PBS supplemented with 0.05% Tween (PBS-T). Anti-cotton rat IgG conjugated with HRP (CCOT25P; ICS, Portland, OR, USA) was added to the wells and incubated for 1 h. Plates were washed three times with PBS. 100 uL of the substrate was added, and the reaction was stopped after 15 min with stop solution.

### 2.5. Detection of Infectious RSV 

Cotton rats were sacrificed 4 days after the challenge and lung tissues and BAL were obtained to detect infectious RSV. A 0.1 mg aliquot of serial 10-fold dilutions of lung homogenates was placed on HEp-2 cells and incubated at 37 °C with shaking every 20 min for 4 h. MEM supplemented with 5% FBS with 0.5% agar was overlaid. Plaque numbers were counted after incubation at 37 °C for 6 days and infectivity was expressed as the number of plaques adjusted to 100 mg of lung tissue [25]. Serial 10-fold dilutions of BAL were placed on HEp-2 cells in a 96-well plate. Cells were incubated at 37 °C for 6 days and infectious virus titers were detected as TCID_50_.

TaqMan PCR in the RSV NP genome region was performed to detect the RSV copy numbers [25].

### 2.6. HE Stains and Immunostaining of RSV

Lungs were expanded to their normal volumes with 4% formaldehyde and submerged in formaldehyde for overnight fixation. The fixed tissue was embedded in paraffin, sectioned, and stained with hematoxylin-eosin (HE). Immunostaining was performed using a four-clone blend of monoclonal antibodies against RSV P, F, and NP (AdB Serotec, Kidlington, UK), and anti-mouse IgG conjugated with HRP (Dako Japan, Tokyo, Japan).

### 2.7. Statistical Analysis

Serum antibody levels after the vaccination and the results of the flow cytometry assay were statistically analyzed for significance by the Mann-Whitney *U* test using SAS Software Release 9.2 for Windows, and significance was defined as *p* < 0.05.

## 3. Results 

### 3.1. Virus Growth and the Development of PA Antibodies

Infectious recombinant viruses were recovered from recombinant cDNA and virus growth was examined. The results are shown in Figure 2A. There was no significant difference in viral growth for the MVAIK, MVAIK/RSA/M2-1, and MVAIK/RSV/NP. The growth curve of MVAIK/RSV/F was the similar to that of the MVAIK [24]. 

Development of PA antibodies after immunization with MVAIK, MVAIK/RSV F, MVAIK/RSA/M2-1, and MVAIK/RSV/NP were previously examined, and no significant difference was reported (Figure 2B) [25]. 

### 3.2. Detection of PA Antibodies Against Measles Virus and NA Against RSV 

In the present study, cotton rats were immunized with more than 10 ^5^ TCID_50_ of recombinant measles virus to induce sufficient immune responses. PA antibody titers are an appropriate index for identifying responses to measles virus-based vaccines. PA antibodies were detected on Day 21 (Figure 3A), and increased up to 10 × 2^4.6^, 10 × 2^5.0^, 10 × 2^7.0^, and 10 × 2^7.0^ in the F 10^^5^, F 10^^6^, F + NP, and F + M2-1 groups, respectively, on Day 56 before re-immunization. Cotton rats were re-immunized on Day 56, and PA antibody titers increased up to 10 × 2^6.3 ± 0.33^, 10 × 2^7.0 ± 0.67^, 10 × 2^8.0^, and 10 × 2^8.0^ in the F 10^^5^, F 10^^6^, F + NP, and F + M2-1 groups, respectively, on Day 84. In the MVAIK group, PA antibody titers did not increase in one of the three cotton rats, and this cotton rat did not respond to the vaccine after re-immunization. The reason for the negative response was not explained; however, non-responders are observed at a very low incidence in clinical settings in humans.

IgG EIA antibodies against RSV were detected in sera using ELISA (Figure 3B). The results obtained are shown as absorbance data. Anti-RSV IgG EIA antibodies were detected after Day 35 in the F 10^^5^ group. In the F 10^^6^, F + NP, and F + M2-1 groups, IgG EIA antibodies increased after re-immunization. IgG EIA became positive in MVAIK group on Day 35 but decreased afterward. It was considered false positive because of no detectable NA against RSV.

The results of NA titers against RSV subgroup A are shown in Figure 3C. In the F 10^^5^, F + NP, and F + M2-1 groups, NA titers increased to 2^3.32 ± 0.00^ on Day 21. NA titers increased to 2^4.66 ± 0.33^ and 2^5.32 ± 0.33^ before re-immunization in the F 10^^5^ and F 10^^6^ groups. In the F + NP and F + M2-1 groups, NA titers increased to 2^5.32 ± 0.00^ on Day 84. Regarding IgG EIA data, the F 10^^5^ group had the highest values (Figure 3B), whereas the F 10^^6^ group had the highest NA titers (Figure 3C). No NA titer against RSV was consistently detected in the naïve or MVAIK groups. BAL was collected on Day 88 when animals were sacrificed after the RSV challenge. The results of anti-RSV IgG EIA antibodies in BAL obtained on Day 88 are shown in Figure 3D. Higher levels of RSV EIA antibodies were detected in BAL in the F 10^^5^, F 10^^6^, F + NP, and F + M2-1 groups than those from the naïve group, but not significant (*p* = 0.07) (Figure 3D).

### 3.3. RSV-Specific CD8^+^/IFN-γ^+^ Cells in the Spleen

Spleen cells were obtained after the RSV challenge in each group and stimulated with six peptides as listed in Table 1. These peptides were described previously [26,27,28]. The percentage of RSV-specific CD8^+^ cells in the spleen was counted and the results obtained are shown in Figure 4. No response was observed in naïve and MVAIK empty vector groups. The proportion of CD8^+^/IFN-γ^+^ cells increased by 1.97 ± 0.27-fold (NP_306–314_) and 2.20 ± 0.35-fold (NP_360–368_) following a stimulation with NP peptides in the F + NP group. RSV-F-specific CD8^+^/IFN-γ^+^ cells were similarly induced in the F+NP group, whereas the response of RSV-F-specific CD8^+^ cells was weaker in the F 10^5 and F 10^6 groups. CD8^+^/IFN-γ^+^ responses to RSV-M2-1 were weak, and we supposed that M2-1 epitope regions have not been identified in cotton rats.

### 3.4. Analysis of CD4^+^/IFN-γ^+^ and CD8^+^/IFN-γ^+^ Cells in the Lung 

Cells were collected from the homogenate of the lung tissues after the RSV challenge and fixed without stimulation because lymphocytes were considered to respond to RSV after the challenge. The results are shown in Figure 5. The number of CD8^+^/IFN-γ^+^ cells was 16,099 ± 6525 per 10^5^ cells in the MVAIK group and no significant difference was observed in comparison with that in the naïve group. The numbers of CD8^+^/IFN-γ^+^ cells were 9933 ± 5898 and 7389 ± 1441 per 10^5^ cells in the F 10^5 and F 10^6 groups, respectively. Those in the F 10^5 and F 10^6 groups were significantly lower than that in the naïve group. In the F+NP group, the number of CD8^+^/IFN-γ^+^ cells was 46,874 ± 1631, which was significantly higher than that in the naïve and MVAIK groups. The number of CD8^+^/IFN-γ^+^ cells was 44,852 ± 6501 in the F+M2-1 group, significantly higher, compared with the MVAIK group. On the other hand, CD4^+^/IFN-γ^+^ cells were induced in all immunized groups and their numbers were significantly higher in the F+NP and F+M2-1 groups than in the other groups (*p* < 0.01). 

### 3.5. Histopathological Scoring 

In our previous study, inflammatory cells were infiltrated into the lungs of immunized animals with MVAIK/RSV/F after the challenge, and mild pneumonitis was observed. In the naïve and MVAIK groups, severe inflammation was demonstrated: increased thickness of alveolar wall, mucus in the alveolar space, and destruction of bronchial epithelial cells. Cotton rats in the F 10^^5^ and F 10^^6^ groups developed mild pneumonitis with the infiltration of inflammatory cells after the challenge (Figure 6A). Pneumonitis was absent in the F + M2-1 and F + NP groups. RSV antigens were not detected in the lungs of cotton rats immunized with recombinant measles viruses. Histological scores decreased in the groups immunized with recombinant measles viruses (Figure 6B). Scores were lower in the F + NP and F + M2-1 groups than in the F 10^^5^ and F 10^^6^ groups. These results suggest that cellular immune responses contributed to better histopathological scores. 

### 3.6. Recovery of Infectious Virus after the Challenge

Infectious viruses were recovered from the naïve and MVAIK groups, but not from the F 10^^5^, F 10^^6^, F+M2-1, and F+NP groups after the challenge (Figure 7A). The copy numbers of the RSV-NP genome were investigated in lung homogenates and were detected in the naïve and F+M2-1 groups (Figure 7B). 

## 4. Discussion

NA against RSV was induced in cotton rats immunized with MVAIK/RSV/F [24], and strong RSV-specific CTL activities were induced in those immunized with MVAIK/RSV/NP and MVAIK/RSV/M2-1 [25]. EIA antibodies specific to RSV proteins were detected in the F+NP and F+M2-1 groups. The expression of RSV-NP and M2-1 proteins by recombinant measles viruses was analyzed in culture medium using SDS-PAGE, and they were confirmed not to be released [25]. RSV-NP and M2-1 are intracellular proteins and not transported to extracellular spaces. These proteins were released through cytopathic effects of measles virus or cytolysis by the cellular immune responses of cytotoxicity. The RSV-M2-1 protein functions as both an elongation factor and anti-termination factor in cells [29,30,31]. RSV-NP binds tightly with the viral RNA genome and provides protection against RNase [32]. The RSV-NP and M2-1 proteins were not involved in the attachment of infection process by RSV, and antibodies against RSV-NP and M2-1 proteins did not exhibit NA activities. 

RSV F protein is an envelope protein and works in virus fusion. Native F protein is cleaved into F1 and F2 subcomponents and F1 subcomponent forms stable intermolecular binding between two heptads repeats and forming a trimer. During fusion process, the F protein induces its conformational changes (post-fusion) [22]. Most of the neutralizing activity is directed against the F protein and recently several antigenic epitopes are determined in the pre- and post-fusion conformations by monoclonal antibodies [33]. Neutralizing activity decreased dramatically after the adsorption of human sera with pre-fusion protein, but adsorption with post-fusion protein removed approximately 30% of neutralizing activity [34]. The effective vaccine approaches should aim to use pre-fusion antigens, and the present recombinant MVAIK/RSV/F is likely to express the native F protein. However, further work is required to determine the structure of the RSV F protein expressed by vector-based vaccine.

Therefore, antibodies against RSV-F protein work as NA against RSV in the F+NP and F+M2-1 groups. In these groups, the dose of infectivity of MVAIK/RSV/F was same as the F 10^^5^ group and lower than the F 10^^6^ group. Therefore, higher NA titers were noted in the F 10^^6^ group. Although secreted IgA antibodies were not examined in the present study, we demonstrated that cotton rat IgG EIA antibody against RSV was detected in BAL (Figure 3D). Virus-specific IgG in BAL was reported in HIV-infected patients, and IgG levels in lung lavage reflected serum IgG values but did not correlate with lung immunoglobulin-producing cells [35,36]. Local IgG antibody was considered the exudates from serum [35]. We also analyzed the relationship between IgG levels in serum and BAL, and found no correlation (*R*^2^ = 0.27, data not shown). We could not examine the NA titers in BAL but, however, virus-specific IgG in BAL contain NA antibodies, which may inhibit viral infection in the lower respiratory tract. IgG in BAL may be equally effective as secreted IgA in the local protection of bronchial epithelial cells.

Inflammatory cells infiltrated into the lungs and induced severe pneumonitis in the FI-RSV trial following natural infection with RSV [8,9]. In our previous study, cotton rats immunized with only MVAIK/RSV/F showed inflammatory cells around the bronchial spaces after the challenge, and this was confirmed in the present study. This finding led to the hypothesis that immune responses induced by MVAIK/RSV/F are insufficient to activate cellular immune responses [24]. Inflammatory responses after the RSV challenge were inhibited by CD8^+^ cells (CTL) and Th1-type cytokines, such as IFN-γ [37,38]. The mechanisms underlying the inhibition of inflammatory responses after the RSV challenge involved the down-regulation of Th2-type cytokine secretion by CTL and Th1-type cytokines [39]. In the present study, CTL responses were reinforced using simultaneous immunization with MVAIK/RSV/NP and MVAIK/RSV/M2-1 (F + M2-1 and F + NP groups). In the F+NP group, the induction of RSV-NP-specific CD8^+^/IFN-γ^+^ cells was demonstrated in the spleen. On the other hand, RSV-M2-1-specific CD8^+^/IFN-γ^+^ cells were not detected in the F+M2-1 group stimulated with M2-1 peptides. M2-1 epitope regions have not been identified in cotton rats. Two M2-1 peptides were used in the present study, and the number of RSV-M2-1-specific CD8^+^/IFN-γ^+^ cells was low in the F + M2-1 group. In a previous study, mouse dominant and subdominant epitope regions were used; however, RSV-M2-1-specific CD8^+^/IFN-γ^+^ cells were not detected similar to the present study. M2-1 epitopes recognized by cotton rat may differ from those reported in the mouse model.

Local immune responses against RSV were induced after immunization because CD4^+^/IFN-γ^+^ and CD8^+^/IFN-γ^+^ cells were induced in the lungs obtained from the cotton rats immunized in F+NP and F+M2-1 groups, and no infectious viruses were detected after the challenge. Unexpectedly, the number of CD8^+^/IFN-γ^+^ cells was low in the lung of cotton rats immunized with MVAIK/RSV/F alone after RSV challenge, similar to the empty vector (MVAIK) group. The lower response after RSV challenge in F 10^^5^ and F 10^^6^ groups may indicate the lower CTL response probably because of transient T cell suppression after immunization with MVAIK. But, the expression of NP and M2-1 enhanced CD8^+^/IFN-γ^+^ responses.

These results indicate that cotton rats developed both humoral and cellular immunities, and viruses were cleared without severe pneumonitis (Figure 7A). Although IFN-γ induction was observed in the F 10^^5^ and F 10^^6^ groups, infiltration of inflammatory cells was noted around the bronchial space. These animals also showed the induction of CD4^+^/IFN-γ^+^ cells in the lungs, compared with naïve and MVAIK groups; however, the number of CD8^+^/IFN-γ^+^ cells was significantly lower than that in the control group after the RSV challenge. These results indicate that RSV-F-induced IFN-γ and RSV-specific CD4^+^ T cells were dominantly activated and expressed IFN-γ in the lungs after RSV infection. RSV F protein is known to induce Th1 response, but internal proteins are rich in T-cell epitopes. NP or M2-1 protein was expressed and released during the replication of recombinant measles vaccine and induced effective T-cell mediated immune responses [40]. 

Standard potency of measles AIK-C strain induced sufficient serological responses in infants aged 4–5 months in Togo [41] and in those aged six months in clinical trials conducted in several countries [42]. But, the peak of hospitalization of RSV infection occurs <3 months of age, and the measles virus-vectored vaccine cannot cover this population. Another target is elderly with lower antibody titers against measles virus. Measles virus-vectored vaccine would replicate and provide immunity against RSV.

## 5. Conclusions

In the present study, combined immunization with MVAIK/RSV/F, MVAIK/RSV/NP, or MVAIK/RSV/M2-1 induced NA and CTL responses. RSV-specific CD8^+^IFN-γ^+^ cell numbers increased after re-immunization without inflammatory responses in the lungs after the RSV challenge. 

## Figures and Tables

**Figure 1 vaccines-07-00027-f001:**
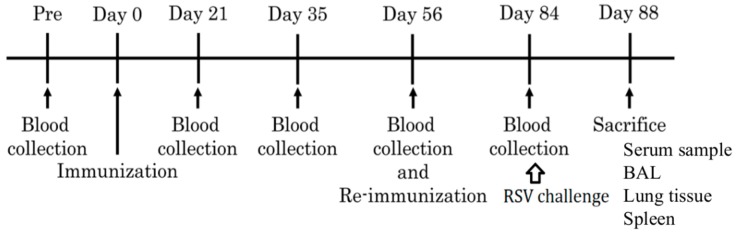
Schedule for immunization, the challenge test, and sample collection. Recombinant vaccines were administered on Days 0 and 56 and cotton rats were challenged with RSV on Day 84. They were sacrificed four days after the RSV challenge (Day 88).

**Figure 2 vaccines-07-00027-f002:**
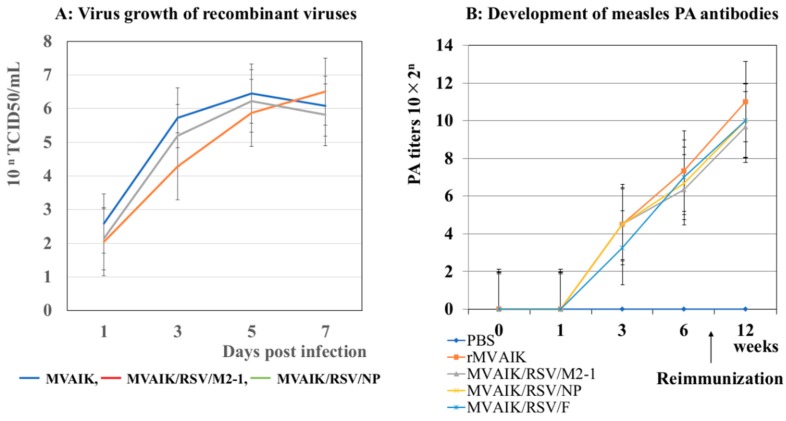
Virus growth and the development of PA antibodies. (**A**) Virus growth of the recombinant MVAIK, MVAIK/RSV/NP, and MVAIK/RSV/M2-1. Culture fluids were obtained on Days 1, 3, 5, and 7 of the culture and infectivity was shown as 10^n^TCID50 in Vero cells. (**B**) Development of PA antibodies. Cotton rats were immunized with MVAIK, MVAIK/RSV/F, MVAIK/RSV/NP, and MVAIK/RSV/M2-1, and re-immunized at 8 weeks after the first dose. Blood samples were obtained serially, and PA antibody was examined. Figure 2B was reported in reference No. 25.

**Figure 3 vaccines-07-00027-f003:**
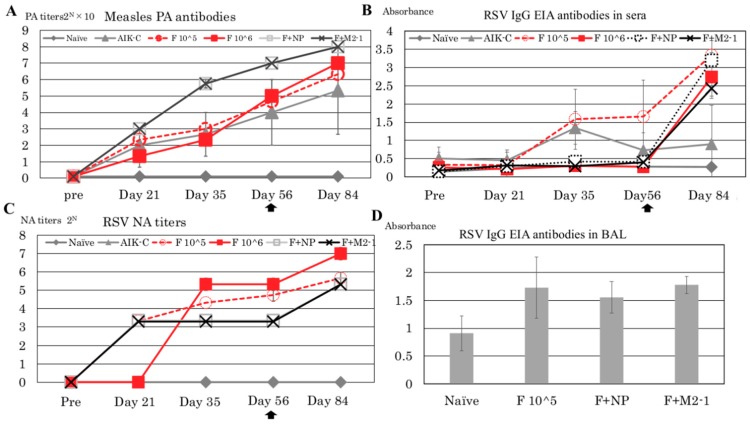
Serological responses against measles virus and RSV. (**A**): PA antibodies against measles virus in serum samples from cotton rats are shown in different groups collected before immunization and on Days 21, 35, 56, and 84. The second dose of recombinant virus was given on Day 56. (**B**) Serum IgG EIA antibodies against RSV are shown by absorbance at 450 nm. (**C**) NA titers against RSV are shown as 2^n^ of the dilution of serum samples by a 50% plaque reduction. (**D**) IgG EIA antibodies in BAL obtained after the RSV challenge (Day 88) are shown by absorbance at 450 nm. BAL was diluted at 1:20. The bar represents the means of 3~4 rats per group ± SEM.

**Figure 4 vaccines-07-00027-f004:**
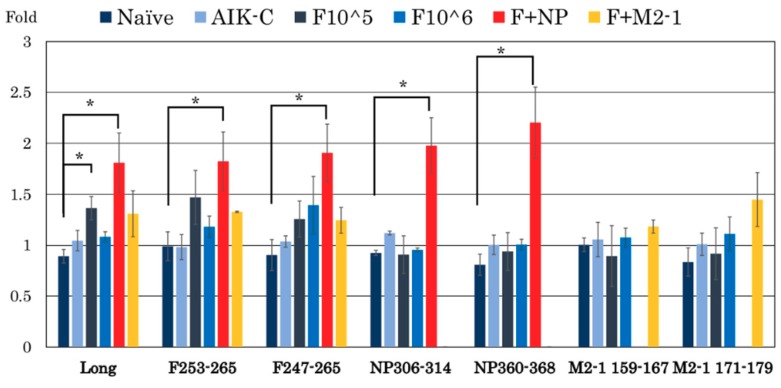
The proportion of CD8^+^/IFN-γ^+^ cells in spleen cells obtained four days after the RSV challenge (Day 88), stimulated with UV-inactivated RSV Long strain, F, NP, and M2-1 peptides. Spleen cells were obtained after the RSV challenge and analyzed for IFN-γ production by CD8^+^ cells. The bar represents the mean of 3~4 rats per group ± SEM. (* *p* < 0.05).

**Figure 5 vaccines-07-00027-f005:**
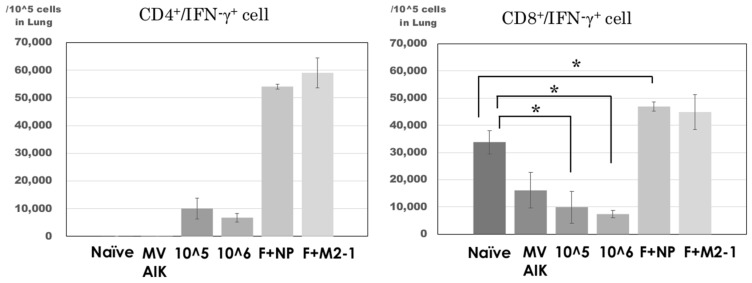
Detection of IFN-γ^+^ expressing cells in lung homogenates. Lung homogenates were obtained after the RSV challenge on Day 88. IFN-γ-producing cells were separated further into CD8^+^ and CD4^+^ cells. The bar represents the mean of 3~4 rats per group ± SEM. (* *p* < 0.05).

**Figure 6 vaccines-07-00027-f006:**
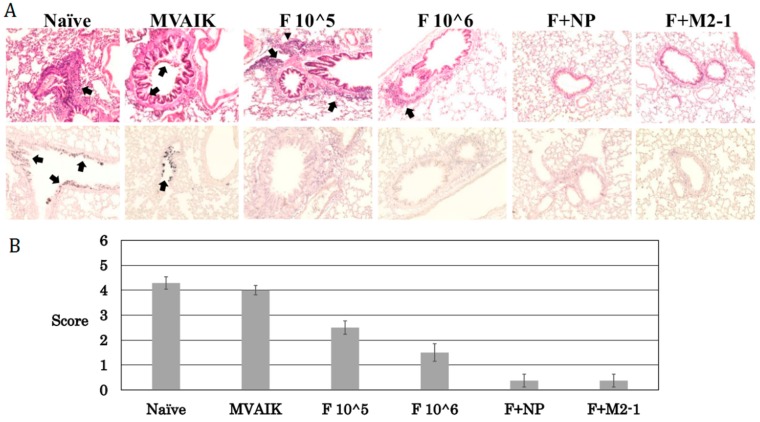
Histopathological findings and scoring. (**A**) Lung tissues were collected after the RSV challenge on Day 88. The upper panels show the results of HE stains, and arrows indicate the pathological findings as scoring. The lower panels are immunostaining and arrows indicate the positive staining of RSV. (B) Severity of inflammatory response was evaluated by scoring the microscopic findings of an average score in eight different sections for each group, using the following criteria: 1, increased thickness of alveolar walls; 1, destruction of bronchial epithelial cells; 1, peri-bronchial infiltration of inflammatory cells; 2, mucus as an exact cause of occlusion in the bronchial space; and 1, bleeding. Microscopic images were taken with a Life Technologies EVOS XL Core light microscope at 40× magnification.

**Figure 7 vaccines-07-00027-f007:**
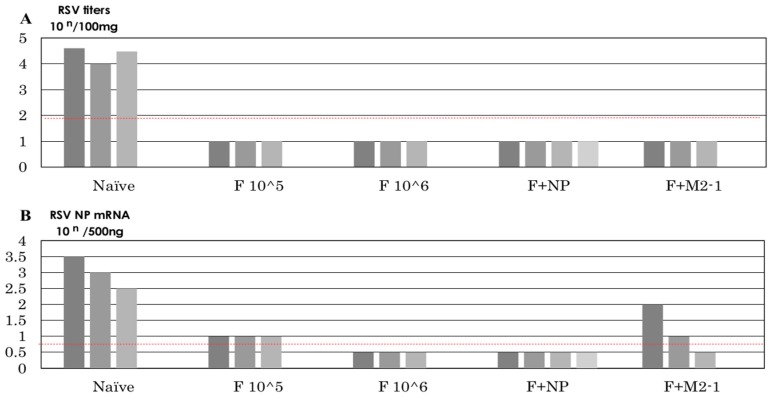
Recovery of infectious RSV and genome copy numbers after the RSV challenge. (**A**) Lung tissues were homogenized and diluted. Lung homogenates were placed on HEp-2 cells and incubated for six days. Plaque numbers were counted. Infectivity is shown as PFU in 100 mg of lung tissue. (**B**) The copy number of the RSV-NP gene is shown. TaqMan PCR was performed in the RSV-NP gene region. Dotted red lines indicate the detection limits.

**Table 1 vaccines-07-00027-t001:** List of peptides for RSV-F, NP, and M2-1.

Peptides	Sequence	MHC
F_247–261_	VSTYMLTNSELLSLI	unknown
F_253–265_	TNSELLSLINDMP	unknown
NP_306–314_	NPKASLLSL	HLA-B7 (Human)
NP_360–368_	NGVINYSVL	H-2b (Mouse C57BL/6)
M2-1_159–167_	KTIKNTLDI	H-2b (Mouse C57BL/6)
M2-1_171__–__179_	ITINNPKEL	H-2b (Mouse C57BL/6)

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
