# Peer review of "Simultaneous Administration of Recombinant Measles Viruses Expressing Respiratory Syncytial Virus Fusion (F) and Nucleo (N) Proteins Induced Humoral and Cellular Immune Responses in Cotton Rats"

_vaccines, 2019, doi:10.3390/vaccines7010027_

Round 1

Reviewer 1 Report

This paper describes an RSV F + N protein measles virus vectored candidate vaccine. They evaluate the F alone or F +N vectored vaccine’s effect on virus replication and lung inflammation in cotton rats. They suggest that their data shows the importance of cellular immunity.

1.   I find the introduction confusing with specific ideas, findings, etc. noted but not in a cohesive fashion.  Some findings from the literature appear to be accepted as confirmed but, I believe, many are still just findings and their actual role in disease/immunity is yet to be determined.  On p. 2, 2nd #, the authors state that “Therefore, M2-1 is considered to be necessary for the induction of CTL activities in mice”.  I believe other proteins also induce CTL without M2-1 though M2-1 may be improve the CTL response.

2.   The use MVAIK measles virus as their vaccine vector but do give not details about the virus.  They should provide background information on MVAIK.

3.   They state that their challenge virus (RSV Long) is grown in HEp-2 cells with 10% FCS.  FCS adds considerable antigenic material that induced immune responses that affected results in some studies.  It would be important to include mock-infected cell material to account for responses induced by non-RSV antigens. For example, this control provides assurance that responses to UV-inactivated RSV are specific to RSV and not to other antigens in the preparation.

4.   What is the specificity of the PA test for measles antibody responses?  What measles virus antigens coat the particles?

5.   The figures are faint and should be of better quality. For example, I am not certain which lines in Figure 1 represent which vaccination regimen.

6.   They give the MHC for the peptides for protein-specific stimulation of spleen cells.  Are these cotton rat, human, or other MHC epitopes? Is it correct that the cotton rats are out-bred and response to MHC epitopes will vary among the animals?  If true, this would affect consistency of results and confidence in differences seen among vaccine groups.

7.   Figure 5, shows a decrease compared to naïve animals for CD8 IFN-g+ cells with MVAIK F at both inoculum while the F+NP and F+M2-1 show an increase.  Do the investigators have thought on why this might be?

8.   Figure 6 needs additional explanation. The labels for vaccine group are small and hard to see clearly. The stain/preparation for the two rows is not given. What are the arrows indicating?

9.   Were the experiments repeated or done once?

Author Response

Response to the reviewer 1

Thank you for your valuable comments. Revised portion was written in red letters and followings are my responses.

1.   I find the introduction confusing with specific ideas, findings, etc. noted but not in a cohesive fashion.  Some findings from the literature appear to be accepted as confirmed but, I believe, many are still just findings and their actual role in disease/immunity is yet to be determined.  On p. 2, 2nd #, the authors state that “Therefore, M2-1 is considered to be necessary for the induction of CTL activities in mice”.  I believe other proteins also induce CTL without M2-1 though M2-1 may be improve the CTL response.

Response】 I agree the comment and the sentence was changes: Therefore, NP or M2-1 is considered one of the effective proteins inducing CTL activities in mice [18, 19]. (line 73-74)

2.  The use MVAIK measles virus as their vaccine vector but do give not details about the virus.  They should provide background information on MVAIK.

Response I added the background information in the Introduction as following:

A live-attenuated virus vaccine can induce both humoral and cellular immune responses but is not developed for RSV. Several approaches have been proposed such as replication-defective vectors or chimeric live vectored-vaccine candidates, using parainfluenza virus, Sendai virus, small pox vaccine, and BCG [20]. Most vector viruses except small pox had no experience in human clinical use. The AIK-C measles vaccine strain has been used as National Immunization Program in Japan since 1978 and the safety and immunogenicity are established. Infectious cDNA clone was constructed from the AIK-C seed virus by reverse genetics and the development of recombinant vectored-vaccines derived from licensed vaccine strain is considered to be theoretically safe and effective [21]. (line 75-84)

3.   They state that their challenge virus (RSV Long) is grown in HEp-2 cells with 10% FCS.  FCS adds considerable antigenic material that induced immune responses that affected results in some studies.  It would be important to include mock-infected cell material to account for responses induced by non-RSV antigens. For example, this control provides assurance that responses to UV-inactivated RSV are specific to RSV and not to other antigens in the preparation.

Response I understand the comment and Figure 3 in the previous version was deleted. It was not informative as the reviewer suggested. I mentioned on the results obtained from the stimulation with peptides

4.   What is the specificity of the PA test for measles antibody responses?  What measles virus antigens coat the particles?

Response The PA test is to detect the binding antibody to the measles surface antigens. And following sentence was added: Gelatin particles are coated with purified measles virus antigens. (line 180)

5.   The figures are faint and should be of better quality. For example, I am not certain which lines in Figure 1 represent which vaccination regimen.

Response Figure 1 demonstrated the immunization schedule and timing for the sampling for all groups (Naïve, MVAIK, F 10^5, F 10^6, F+NP and F+M2-1).

6.   They give the MHC for the peptides for protein-specific stimulation of spleen cells.  Are these cotton rat, human, or other MHC epitopes? Is it correct that the cotton rats are out-bred and response to MHC epitopes will vary among the animals?  If true, this would affect consistency of results and confidence in differences seen among vaccine groups.

Response Cotton rats were inbred in our laboratory for more than 7-8 years. And it was added as following: The experimental scheme is shown in Figure 1. Seven- to eight-week-old female inbred cotton rats (three or four animals per group) were immunized……..

7.   Figure 5, shows a decrease compared to naïve animals for CD8 IFN-g+ cells with MVAIK F at both inoculum while the F+NP and F+M2-1 show an increase.  Do the investigators have thought on why this might be?

Response RSV F protein is known to induce Th1 responses but internal proteins are rich in T-cell epitopes. NP or M2-1 protein was expressed and released during the replication of recombinant measles vaccine and induced effective T-cell mediated immune responses [38]. (line 416-418)

8.   Figure 6 needs additional explanation. The labels for vaccine group are small and hard to see clearly. The stain/preparation for the two rows is not given. What are the arrows indicating?

Response I revised the figure legend of Figure 5.

Figure 5. Histopathological findings and scoring.

A: Lung tissues were collected after the RSV challenge on Day 88. The upper panels show the HE staining and arrows indicate the pathological findings as scoring. The lower panels are immunostaining and arrows indicate the positive staining. B: Regarding the scoring of histological findings, eight random fields per group were scored for histopathology using the following criteria: 1, increased thickness of alveolar walls; 1, destruction of bronchial epithelial cells; 1, peri-bronchial infiltration of inflammatory cells; 2, mucus as an exact cause of occlusion in the bronchial space; and 1 bleeding. Microscopic images were taken with a Life Technologies EVOS XL Core light microscope at 40× magnification. (line 314-323)

9.   Were the experiments repeated or done once?

Response We did this experiments twice, and the experiments of MVAIK/RSV/F, MVAIK/RSV/NP, and MVAIK/RSV/M2-1 were done in several times.

Reviewer 2 Report

The paper by Yamaji et al examines the immune response of recombinant measles viruses expressing RSV F, N, or M2-1 proteins. Previous work showed that these vaccines, independently, provide insufficient or unsafe protection against challenge with RSV. Here, combinations of F and N expressing vaccines or F and M2-1 expressing vaccines are used to examine whether safe and more comprehensive immunity can be achieved. The authors demonstrate that a combination of measles viruses separately expressing F and N, yields the highest level of RSV-specific IFNgamma-secreting CD8 T cells in cotton rats, lowers histopathological scores, and lowers viral load in the lungs after challenge with RSV. Two intramuscular doses of vaccine are necessary to achieve the described effects.

Although the combined vaccines induce a better response than the individual vaccines, there is no context that helps the reader understand how the improved vaccines compare to other vaccines in the field. In particular, there is no information on the conformation of F. In the RSV field, it is now well established that the regular F protein constitutes a poor antigen as it induces a high proportion of postfusion F antibodies, which are much less neutralizing than prefusion F antibodies and can induce disease related to the enhanced disease found in a previous trial with inactivated vaccine. In addition, there are a number of weaknesses/concerns that lower the impact of the work. The manuscript presents a small increment of data relative to previous work, and often has insufficient description of why the data were pursued and how they were interpreted. Almost all figure legends lack important information, and little interpretation is provided to explain or discuss the data, making it difficult for the reader to interpret the findings.

Additional more specific comments:

- The paper needs characterization of RSV protein expression by the different recombinant viruses, or clear references as to the levels of RSV antigens made by these viruses in previously published work. The only information present says that some RSV antigens were not released (line 263); it is unclear why they were expected to be released, since they are all internal or membrane-anchored proteins.

- There is a general scarcity of information which makes it diffcult to interpret the findings. For example, it is not described what MVAIK is, and there is no statement anywhere that MVAIK-C is used as a control (without RSV genes). Some abbreviations are not explained. It is not explained why two different amounts of the F virus are used, but only single amounts for the other viruses. The rationales for experiments are sometimes unclear or inadequate. Line 161, where the authors state they looked at IgG, without explaining the relevance of serum or lung IgG for RSV disease. Most figure legends lack critical information. Fig 3A does not show where the CD8 T cells were harvested. Fig 4 legend does not show at which time in the experiment the indicated cells were harvested. Fig 5 legend or corresponding text does not state at which time the T cells were harvested. Etc.

- Fig 2A and C: the data points of viruses F+N and F+M2-1 are 100% the same at every timepoint? This seems highly unlikely.

- Fig 2A. MVAIK/RSV/F+NP shows much higher PA antibodies than the others. What does that mean? Did this virus perhaps replicate better than the others? There is no discussion or controls or references to previous work to show that these viruses replicate (or not) to equivalent levels.

- Fig 2B: negative control virus AIK-C shows RSV IgG antibodies in sera, and 10ˆ5 PFU of F vaccine gives higher titer RSV IgG than 10ˆ6 PFU. These unexpected findings are not discussed.

- Fig 2C: neutralization titers are higher for the F virus (10ˆ6) than for the F+NP or F+M2-1 virus. Please discuss.

- Fig 2D: The importance or relevance of IgG in BAL is not well explained. In the discussion, lung IgG level in HIV patients are referenced, but it is not clear what the implications are. In line 273, it is stated that ‘IgG in BAL indicates the prevention of pneumonitis caused by viral infection’. How the authors derive this indication from the data is unclear.

- Fig 2D. Authors claim more IgG for all vaccines; there is no p value to confirm this.

- Fig 3B does not appear to add any information and may not be appropriate for the question asked. As the authors say, there may not be any M2-1 CD8 targets in cotton rats. In addition, the N protein is internal in UV-inactivated particles. Fig 4 compares CD8 specificities by stimulations by peptides which is more appropriate and informative.

- The last line of the conclusion (‘protection was required for immune responses…’) is unclear. What are the authors trying to say?

- Line 270 says that antibodies against RSV-G protein were neutralizing? RSV G is not included in the vaccines.

- Fig 4 and 5. In lung homogenates, the levels of IFNgamma expressing T cells varies much more between the different viruses than in the spleen. This difference should be discussed.

- Fig 5. Measles virus vaccination appears to lower the number of IFNgamma expressing T cells in the lung. What does that mean for measles virus as a vector (since induction of CD8 Tcells is important for protection against RSV)?

Author Response

Response to the Reviewer 2

Thank you for your comments. The followings are my response and revised portions are written in red letters. I added 4 references to discuss the comments.

Although the combined vaccines induce a better response than the individual vaccines, there is no context that helps the reader understand how the improved vaccines compare to other vaccines in the field. In particular, there is no information on the conformation of F. In the RSV field, it is now well established that the regular F protein constitutes a poor antigen as it induces a high proportion of post-fusion F antibodies, which are much less neutralizing than prefusion F antibodies and can induce disease related to the enhanced disease found in a previous trial with inactivated vaccine. In addition, there are a number of weaknesses/concerns that lower the impact of the work. The manuscript presents a small increment of data relative to previous work, and often has insufficient description of why the data were pursued and how they were interpreted. Almost all figure legends lack important information, and little interpretation is provided to explain or discuss the data, making it difficult for the reader to interpret the findings.

Response Regarding the F protein antigen, the following paragraph was added in the Discussion. RSV F protein is an envelope protein and works in virus fusion. Native F protein is cleaved into F1 and F2 subcomponents and F1 subcomponent forms stable intermolecular binding between two heptads repeats, and forming a trimer. During fusion process, the F protein induces its conformational changes (post-fusion) [20]. Most of the neutralizing activity is directed against the F protein and recently several antigenic epitopes are determined in the pre- and post-fusion conformations by monoclonal antibodies [31]. Neutralizing activity decreased dramatically after the adsorption of human sera with pre-fusion protein, but adsorption with post-fusion protein removed approximately 30% of neutralizing activity [32]. The effective vaccine approaches should aim to use pre-fusion antigens, and the present recombinant MVAIK/RSV/F is considered to express the native F protein. (line 357-367)

   We did not examine the F protein conformation, but recent review article mentioned that the pre-fusion F protein antigen could be expressed from a vector, or chimeric virus. This review article was cited as ref. 38, and following sentence was added in Conclusion:

In the present study, combined immunization with MVAIK/RSV/F, MVAIK/RSV/NP, or MVAIK/RSV/M2-1 induced NA and CTL responses. RSV-specific CD8+IFN-γ+ cell numbers increased after re-immunization without inflammatory responses in the lungs after the RSV challenge. The present measles vaccine-vector recombinant viruses express and deliver these antigens, probably pre-fusion F antigen, during replication and is expected as a promising RSV vaccine candidate [38]. (line 420-426) 

1- The paper needs characterization of RSV protein expression by the different recombinant viruses, or clear references as to the levels of RSV antigens made by these viruses in previously published work. The only information present says that some RSV antigens were not released (line 263); it is unclear why they were expected to be released, since they are all internal or membrane-anchored proteins.

Response SV-NP and M2-1 are intracellular proteins, and are not transported to extracellular spaces. These proteins were released through cytopathic effects of measles virus or cytolysis by the cellular immune responses of cytotoxicity. (line 350-351) Full length F gene was cloned and the recombinant measles virus expressed the F protein and released by cytopathic effects.

2- There is a general scarcity of information which makes it difficult to interpret the findings. For example, it is not described what MVAIK is, and there is no statement anywhere that MVAIK-C is used as a control (without RSV genes). Some abbreviations are not explained. It is not explained why two different amounts of the F virus are used, but only single amounts for the other viruses. The rationales for experiments are sometimes unclear or inadequate. Line 161, where the authors state they looked at IgG, without explaining the relevance of serum or lung IgG for RSV disease. Most figure legends lack critical information. Fig 3A does not show where the CD8 T cells were harvested. Fig 4 legend does not show at which time in the experiment the indicated cells were harvested. Fig 5 legend or corresponding text does not state at which time the T cells were harvested. Etc.

Response The following paragraph was added in the Materials and Methods to explain the background:

Full length infectious cDNA was constructed from the AIK-C live attenuated measles vaccine strain, and infectious virus (MVAIK) was recovered. The Asc I restriction site was artificially introduced at the P/M junction and heterologous genes were inserted using the Asc I restriction site [21]. The full-length F gene of wild-type RSV was cloned at the P/M junction and infectious recombinant virus (MVAIK/RSV/F) was recovered [22]. The RSV-M2-1 and NP genomes were cloned from RSV subgroup A wild type and inserted at the P/M junction of the measles virus vector. Recombinant measles viruses (MVAIK/RSV/M2-1 and MVAIK/RSV/NP) were recovered from B95a cells. (line 99-107)

Following sentence was added in 2.2. Animals and immunization

Rats non-immunized (naïve) and immunized with MVAIK (empty vector of MVAIK: AIK-C group) were used as control. (line 128-130)

The reason was mentioned why two different amounts of the F virus are used:

Response Cotton rats were immunized with a total of 5×105 TCID50 of MVAIK/RSV/F and 5×105 TCID50 of MVAIK/RSV/NP in the group F+NP, and those immunized with 5×105 TCID50 of MVAIK/RSV/F and MVAIK/RSV/M2-1 in the F+M2-1 group. In the F10^6, F+NP, and F+M2-1 groups, vaccination dosages were adjusted to the same infectivity of measles virus. (line 119-123)

- Fig 2A and C: the data points of viruses F+N and F+M2-1 are 100% the same at every timepoint? This seems highly unlikely.

- Fig 2A. MVAIK/RSV/F+NP shows much higher PA antibodies than the others. What does that mean? Did this virus perhaps replicate better than the others? There is no discussion or controls or references to previous work to show that these viruses replicate (or not) to equivalent levels.

Response MVAIK/RSV/M2-1, MVAIK/RSV/NP, and MVAIK viruses showed the same virus growth [22].(line 109-110)

MVAIK/RSV/M2-1 and MVAIK/RSV/NP induced T cell immune response of CD4+IFN-γ+ cells. T-cell help might influence the higher antibody responses.

- Fig 2B: negative control virus AIK-C shows RSV IgG antibodies in sera, and 10ˆ5 PFU of F vaccine gives higher titer RSV IgG than 10ˆ6 PFU. These unexpected findings are not discussed.

Response IgG EIA became positive in MVAIK group on day 35, but decreased afterward. It was considered false positive because of no detectable NA against RSV. (line 235-237)

- Fig 2C: neutralization titers are higher for the F virus (10ˆ6) than for the F+NP or F+M2-1 virus. Please discuss.

Response Therefore, antibodies against RSV-F proteins work as NA against RSV in the F+NP and F+M2-1 groups. In these groups, the dose of infectivity of MVAIK/RSV/F was same as the group F 10^5 groups and lower than in F 10^6 group. Therefore, higher NA titers were noted in F 10^6 group. (line 368-371)

- Fig 2D: The importance or relevance of IgG in BAL is not well explained. In the discussion, lung IgG level in HIV patients are referenced, but it is not clear what the implications are. In line 273, it is stated that ‘IgG in BAL indicates the prevention of pneumonitis caused by viral infection’. How the authors derive this indication from the data is unclear.

Response Although secreted IgA antibodies were not examined in the present study,  Local IgG antibody was considered to be the exudates from serum [33]. We also analyzed the relationship between IgG levels in serum and BAL, and found no correlation (R2 =0.27, data not shown). However, virus-specific IgG in BAL may inhibit viral infection in the lower respiratory tract, and IgG in BAL may be equally effective as secreted IgA in the local protection of bronchial epithelial cells.

- Fig 2D. Authors claim more IgG for all vaccines; there is no p value to confirm this.

Response There was no significant increase (p=0.07) and this paragraph was changed as following:

The results of anti-RSV IgG EIA antibodies in BAL obtained on Day 88 are shown in Fig. 2D. Higher levels of RSV EIA antibodies were detected in BAL in the F 10^5, F 10^6, F+NP, and F+M2-1 groups than those from the naïve group, but not significant (p=0.07) (Fig. 2D). (line 245-248)

- Fig 3B does not appear to add any information and may not be appropriate for the question asked. As the authors say, there may not be any M2-1 CD8 targets in cotton rats. In addition, the N protein is internal in UV-inactivated particles. Fig 4 compares CD8 specificities by stimulations by peptides which is more appropriate and informative.

Response I understood and deleted the Fig.3.

- The last line of the conclusion (‘protection was required for immune responses…’) is unclear. What are the authors trying to say?

Response It was changed as following:

In the present study, combined immunization with MVAIK/RSV/F, MVAIK/RSV/NP, or MVAIK/RSV/M2-1 induced NA and CTL responses. RSV-specific CD8+IFN-γ+ cell numbers increased after re-immunization without inflammatory responses in the lungs after the RSV challenge. The present measles vaccine-vector recombinant viruses express and deliver these antigens, probably pre-fusion F antigen, during replication and is expected as a promising RSV vaccine candidate [38]. (line 421-426)

- Line 270 says that antibodies against RSV-G protein were neutralizing? RSV G is not included in the vaccines.

Response RSV-G was not involved in this study and it was deleted.

- Fig 4 and 5. In lung homogenates, the levels of IFN gamma expressing T cells varies much more between the different viruses than in the spleen. This difference should be discussed.

Response RSV F protein is known to induce Th1 responses but internal proteins are rich in T-cell epitopes. NP or M2-1 protein was expressed and released during the replication of recombinant measles vaccine and induced effective T-cell mediated immune responses [38]. (line 415-418)

- Fig 5. Measles virus vaccination appears to lower the number of IFN gamma expressing T cells in the lung. What does that mean for measles virus as a vector (since induction of CD8 Tcells is important for protection against RSV)?

Response Measles vaccine vector is a live vaccine and it stimulates the innate immunity. The activated innate immune system induced lower number of CD8+IFN-γ+ cells, not specific to RSV.   

Round 2

Reviewer 1 Report

Remaining issues:

1.      The authors list the peptides and MHC locus for some.  In what species were these epitopes/peptides defined. From the references (p. 6, line 207), only the F peptides are linked to cotton rats.  If not defined in cotton rats, why were they chosen and uncertain how data from these peptides is helpful.

2.      The paper would benefit from substantial editing to improve clarity.

3.     P. 4, the authors state that, “RSV virus particles were coated in 96-well ELISA plates at 4˚C overnight.”  How were the RSV virus particles prepared?  After coating onto the plate and washing with PBS tween it is not likely that the virus would be intact as virus particles. Is it possible that non-measles, non-RSV antigens in the vaccine prep were also present in the RSV coating the plates for the IgG ELISA.  If so, this would explain the reactivity noted with the control preparation. Including control antigen preparation in the ELISA would account for this.  It also raise the question that some of the “RSV antibodies” are actually against other antigens.

4.     The NA titers are given as 2X. Are the titers this value or 10 times this value since the starting dilution was 1:10.  Assuming the titers are the value indicated, the neutralizing titers are just at or just above the level of detection.

5.     In figure 3, the only response to the NP peptides was with the F+M2-1 vaccine.  If correct, it suggests the NP peptide stimulation must be inducing a non-specific response and raises the question of specificity of the peptide stimulation studies.  

6.     The CD8 INF-g response suggests the two F vaccines inhibited this response, i.e. lower than naïve and MVAIK while the F + NP and F + M2-1 increased the response. Is this expected, or unexpected, finding that may indicate an “immunosuppressive effect” of measles virus.

7.     First #, p. 10, the authors discuss potential role of RSV IgG in BAL in protection, however, they did not convincingly detect RSV IgG antibodies in BAL. This discussion does not seem relevant to their findings.

8.     In the last sentence of the discussion, the authors state that F in their vaccine is “probably pre-fusion F antigen. They provide not data to support this conclusion and the low titer of neutralizing antibodies (assuming the figure is correct regarding neut titers) might suggest their F is post-fusion.  

Author Response

Reviewer 1

Comments and Suggestions for Authors  

Remaining issues:

1.      The authors list the peptides and MHC locus for some.  In what species were these epitopes/peptides defined. From the references (p. 6, line 207), only the F peptides are linked to cotton rats.  If not defined in cotton rats, why were they chosen and uncertain how data from these peptides is helpful.

Response】 Table 1 was revised. F peptides are recognition sites of palivizumab (synagis®). NP306-314 is based on the human experiments and the others are on mouse. (line 149-151)

2.      The paper would benefit from substantial editing to improve clarity.

ResponseI did.

3.     P. 4, the authors state that, “RSV virus particles were coated in 96-well ELISA plates at 4˚C overnight.”  How were the RSV virus particles prepared?  After coating onto the plate and washing with PBS tween it is not likely that the virus would be intact as virus particles. Is it possible that non-measles, non-RSV antigens in the vaccine prep were also present in the RSV coating the plates for the IgG ELISA.  If so, this would explain the reactivity noted with the control preparation. Including control antigen preparation in the ELISA would account for this.  It also raises the question that some of the “RSV antibodies” are against other antigens.

Response RSV antigens were prepared from culture fluids of Vero cells infected with the Long strain and were roughly purified through centrifugation by 6000 rpm to remove cellular debris, containing 10 7 PFU/mL. This was added in 2.4 Serology (line 180-18).

In the preliminary experiment, control antigen was prepared using Vero cells. Absorbance value showed <0.2, and higher absorbance >0.5 was considered RSV specific.

4.     The NA titers are given as 2X. Are the titers this value or 10 times this value since the starting dilution was 1:10. Assuming the titers are the value indicated, the neutralizing titers are just at or just above the level of detection.

R1esponse Serum samples were serially diluted by four-fold, starting from a 1:10 dilution, and mixed with an equal volume of RSV (100 PFU) in MEM at room temperature for 1 hour. We counted the plaque number in each dilution and NA titers were calculated as the reciprocal of serum dilutions showing a 50% reduction in the plaque number.

The following sentence was added: NA titers are expressed as 2n.          (line 171-172)

5.     In figure 3, the only response to the NP peptides was with the F+M2-1 vaccine.  If correct, it suggests the NP peptide stimulation must be inducing a non-specific response and raises the question of specificity of the peptide stimulation studies.

Response  NP peptides responded to in the F+NP group. Gray scale figure 3 caused misleading and each group was distinguished by using different colors. Figure 3 was changed.

6.     The CD8 INF-g response suggests the two F vaccines inhibited this response, i.e. lower than naïve and MVAIK while the F + NP and F + M2-1 increased the response. Is this expected, or unexpected, finding that may indicate an “immunosuppressive effect” of measles virus.

Response It was an unexpected phenomenon. The reviewer suggested the immunosuppressive effects of measles vector virus. We can’t deny the possibility, but RSV challenge was performed 12 weeks after the first dose and 4 weeks after the second doses. The following sentences are added: Unexpectedly, the number of CD8+/IFN-γ+ cells was low in the lung of cotton rats immunized with MVAIK/RSV/F alone after RSV challenge. In the naïve and MVAIK groups, the number of CD8+IFN-γ+ cells reflects the innate immune response after RSV the challenge. The lower response after RSV challenge in two F^5 and F^6 groups may indicate the lower CTL memory in comparison the F+NP and F+M2-1 groups. (line 403-408)

7.     First #, p. 10, the authors discuss potential role of RSV IgG in BAL in protection, however, they did not convincingly detect RSV IgG antibodies in BAL. This discussion does not seem relevant to their findings.

Response It is not relevant to the finding because local serological response is one aspect of serological response, together with cellular immunity induced by the recombinant viruses. This issue was discussed as following: Local IgG antibody was considered the exudates from serum [33]. We also analyzed the relationship between IgG levels in serum and BAL, and found no correlation (R2 =0.27, data not shown). We could not examine the NA titers in BAL but, however, virus-specific IgG in BAL contain NA antibodies, which may inhibit viral infection in the lower respiratory tract. IgG in BAL may be equally effective as secreted IgA in the local protection of bronchial epithelial cells. (line 374-378)

8.     In the last sentence of the discussion, the authors state that F in their vaccine is “probably pre-fusion F antigen. They provide not data to support this conclusion and the low titer of neutralizing antibodies (assuming the figure is correct regarding NA titers) might suggest their F is post-fusion.  

Response I agree the comment and there was no data on the expression of pre-fusion or post-fusion forms and deleted this expression from the conclusion. I added the following sentence: We should examine the structure of the F protein expressed by vectored-based vaccine. (line 363-364)

Reviewer 2 Report

Most of the previous review comments are addressed moderately well though in some cases minimally. There is still not much information on how these viruses compare to other vaccines in the field, and questions regarding expressed protein levels have not been addressed by the authors. The latter point remains important, see comments below. Overall, the paper presents original though incremental data that will be of interest to the field.

It’s possible this is due to some incompatibility. However, the line numbers in the response do not match with the line numbers in the downloaded revised manuscript. This made it difficult to find the comments an see them in context within the paragraphs.

Remaining comments/suggestions

1

Original question

- The paper needs characterization of RSV protein expression by the different recombinant viruses, or clear references as to the levels of RSV antigens made by these viruses in previously published work. The only information present says that some RSV antigens were not released (line 263); it is unclear why they were expected to be released, since they are all internal or membrane-anchored proteins.

Response by authors

SV-NP and M2-1 are intracellular proteins, and are not transported to extracellular spaces. These proteins were released through cytopathic effects of measles virus or cytolysis by the cellular imuune response of cytotoxicity. (line 350-351) Full length F was cloned and the recombinant measles virus expressed the F protein and released by cytopathic effects.

Comment to authors

The release part of this question was answered but not the protein expression part. Additional data or references should be provided about protein expression levels. This is related to the point below.

2

Original question

- Fig 2A. MVAIK/RSV/F+NP shows much higher PA antibodies than the others. What does that mean? Did this virus perhaps replicate better than the others? There is no discussion or controls or references to previous work to show that these viruses replicate (or not) to equivalent levels.

Response by authors

MVAIK/RSV/M2-1, MVAIK/RSV/NP, and MVAIK viruses showed the same virus growth [22].

Comment to authors

Authors have added a reference saying claiming that the used viruses ‘grow’ similarly. However, the cited reference 22 does not have the virus expressing NP, only a virus expressing F and one expressing G? As is, the increased PA antibodies in Fig. 2A suggest this virus replicates better than the others, and this may (partly) explain the fact that it is the best vaccine. Growth characteristics are important in this case, otherwise the outcome of the different vaccines could be explained entirely or partly on different levels of replication. The authors need to show the level of replication of, and protein expression by, these viruses side-by-side. If these viruses were directly compared in previous work, it can be referenced. If the viruses are from different publications, then they should be compared in one experiment.

Author Response

Open Review 2

Comments and Suggestions for Authors

Most of the previous review comments are addressed moderately well though in some cases minimally. There is still not much information on how these viruses compare to other vaccines in the field, and questions regarding expressed protein levels have not been addressed by the authors. The latter point remains important, see comments below. Overall, the paper presents original though incremental data that will be of interest to the field.

It’s possible this is due to some incompatibility. However, the line numbers in the response do not match with the line numbers in the downloaded revised manuscript. This made it difficult to find the comments an see them in context within the paragraphs.

Remaining comments/suggestions

Comment to authors

The release part of this question was answered but not the protein expression part. Additional data or references should be provided about protein expression levels. This is related to the point below.

Original question

- Fig 2A. MVAIK/RSV/F+NP shows much higher PA antibodies than the others. What does that mean? Did this virus perhaps replicate better than the others? There is no discussion or controls or references to previous work to show that these viruses replicate (or not) to equivalent levels.

Response by authors

MVAIK/RSV/M2-1, MVAIK/RSV/NP, and MVAIK viruses showed the same virus growth [22].

Comment to authors

Authors have added a reference saying claiming that the used viruses ‘grow’ similarly. However, the cited reference 22 does not have the virus expressing NP, only a virus expressing F and one expressing G? As is, the increased PA antibodies in Fig. 2A suggest this virus replicates better than the others, and this may (partly) explain the fact that it is the best vaccine. Growth characteristics are important in this case, otherwise the outcome of the different vaccines could be explained entirely or partly on different levels of replication. The authors need to show the level of replication of, and protein expression by, these viruses side-by-side. If these viruses were directly compared in previous work, it can be referenced. If the viruses are from different publications, then they should be compared in one experiment.

Response I agree the comments and supplemental figure was added. The expression of the F protein was mentioned in the previous report [22] and that of the NP and M2-1 was also reported [23]. The growth curve of MVAIK/RSV/F showed the similar to that of the MVAIK [22]. MVAIK/RSV/NP and MVAIK/RSV/M2-1 showed similar growth curve as MVAIK and they induced approximately similar PA antibodies when inoculated into cotton rats, shown in the supplemental figure. (line 107-112)

Round 3

Reviewer 1 Report

The authors have responded to all comments with varying completeness.  Additional comments.

1.      The investigators should give the dilution of BAL used for the IgG ELISA, presumably <1:200 dilution used for serum (Figure 2).  It would be helpful to include the specimen dilution in the figure legend.  If the BAL dilution is lower, it likely explains the high absorbance in control animals.  Were the values for the immunized animals significantly above the values for control animals. What were the values for the control immunized animals?  The values are given for the other antibody studies and should be included in the BAL results.

2.      P. 6, 3.2.  The authors should include a control antigen to support their conclusion the responses are RSV-specific.

3.      On p. 6, 3.3 the authors state that “CD4+/IFN-γ+ cells were induced in all immunized groups and their numbers were apparently higher in the F+NP and F+M2-1 groups than in the other groups.”  Were the differences significant?  “Apparently” is not a helpful term.

4.      Comment #7 was not intended to question the relevance of BAL IgG but that I am unconvinced they detected RSV-specific IgG in BAL.

Author Response

Thank you for your comments. Here, I responded to each comment.

Comment 1 The investigators should give the dilution of BAL used for the IgG ELISA, presumably <1:200 dilution used for serum (Figure 2).  It would be helpful to include the specimen dilution in the figure legend.  If the BAL dilution is lower, it likely explains the high absorbance in control animals.  Were the values for the immunized animals significantly above the values for control animals. What were the values for the control immunized animals?  The values are given for the other antibody studies and should be included in the BAL results.

Response】 BAL was diluted at 1:20. This sentence was added in Figure legend (line 271). Absorbance values for the control animals were 0.565, 0.63, and 1.573 in this experiment. EIA values varied according to each individual animal. EIA values obtained from immunized animals after RSV challenge showed higher but not significant.

Comment 2P. 6, 3.2.  The authors should include a control antigen to support their conclusion the responses are RSV-specific.

Response Main purpose of the study is to examine the stimulation with RSV NP, M2-1, and F peptides in line with the previous study (ref. 23). RSV antigen was used for the comparison with the experiments using peptides. A control antigen (culture media of mock infected Vero cells) was not used.

Comment 3 On p. 6, 3.3 the authors state that “CD4+/IFN-γ+ cells were induced in all immunized groups and their numbers were apparently higher in the F+NP and F+M2-1 groups than in the other groups.”  Were the differences significant?  “Apparently” is not a helpful term.

Response I agree the comment. Apparently was deleted and the sentence was changed as following (line 304-306): On the other hand, CD4+/IFN-γ+ cells were induced in all immunized groups and their numbers were significantly higher in the F+NP and F+M2-1 groups than in the other groups (p<0.01).

Comment 4  Comment #7 was not intended to question the relevance of BAL IgG but that I am unconvinced they detected RSV-specific IgG in BAL. 

Response Sample volume of BAL was limited and we could not examine the neutralizing test. And as I mentioned in our response to your comment 1.